# RectiWeather: Photo-Realistic Adverse Weather Removal via Zero-shot Soft Weather Perception and Rectified Flow

## Abstract

Despite significant progress in Adverse Weather Removal (AWR), challenges remain in applying existing methods to real-world scenarios and in generating photo-realistic and visually compelling outcomes. The limited generalization of current approaches can be attributed to their inability to accurately perceive complex degradations in weather-affected images. Moreover, owing to optimization objectives that prioritize distortion losses, discriminative methods often produce overly smooth reconstructions. To address these challenges, we propose **RectiWeather**, a novel AWR framework guided by zero-shot soft perceptions extracted from pre-trained vision–language models (VLMs). Specifically, we design an AWR-specific Question Answering (AWR-QA) module that guides VLMs to produce soft perceptions of weather conditions and low-level attributes. These soft perceptions are then integrated into baseline AWR models through attribute-modulated normalization (AMN) and weather-weighted adapters (WWA), enabling posterior mean estimation while minimizing distortion loss. Furthermore, we map the posterior output to the clean image distribution using a perception-aware rectified flow model, where soft perceptions define the source distribution and guide the velocity field. Extensive experiments show that RectiWeather consistently surpasses state-of-the-art baselines in fidelity and perceptual metrics across both all-in-one and out-of-distribution scenarios. Our code will be released upon publication.

## 1 Introduction

As a fundamental task in computer vision, Adverse Weather Removal (AWR) aims to restore weather-degraded images to clean counterparts, which is essential for emerging sectors such as autonomous driving (Zang et al., 2019). These weather degradations include, but are not limited to, rain (Li et al., 2019; Zhang et al., 2024), snow (Chen et al., 2021; Zhang et al., 2021), haze (Li et al., 2017; Cai et al., 2016; Li et al., 2020a; Song et al., 2023), and low light conditions (Zhou et al., 2024; 2025). To address these weather degradations, many works focus on networks targeting a single weather degradation (Zang et al., 2019; Li et al., 2019; Fu et al., 2017; Zhang et al., 2024; Li et al., 2017; Cai et al., 2016; Li et al., 2020a; Song et al., 2023), while only a few works have proposed non-blind general-purpose models designed to restore all weather conditions (Li et al., 2020b). However, these models typically depend on predefined priors or separate task-specific models for each type of weather degradation, making them less reliable in blind real-world scenarios. For improved efficacy, recent studies have increasingly focused on all-in-one blind restoration models (Valanarasu et al., 2022; Ye et al., 2023; Yang et al., 2024; Potlapalli et al., 2023; Luo et al., 2023a; Rajagopalan & Patel, 2025), designed to adaptively handle various weather conditions within a single network.

Early blind AWR methods primarily rely on traditional Convolutional Neural Networks (CNNs) (Li et al., 2020b; Zhu et al., 2023; Krizhevsky et al., 2012), which restore degraded images by learning spatial patterns and fine-grained features via convolutional operations. Transformers (Vaswani, 2017) have also been extensively studied for AWR (Potlapalli et al., 2023; Valanarasu et al., 2022; Sun et al., 2024; Wang et al., 2024; Cui et al., 2025), often outperforming traditional CNNs with their superior ability in capturing global dependencies. Diffusion-based approaches (Özdenizci & Legenstein, 2023; Luo et al., 2023a; Zheng et al., 2024) have also been shown to be highly effective in AWR via their

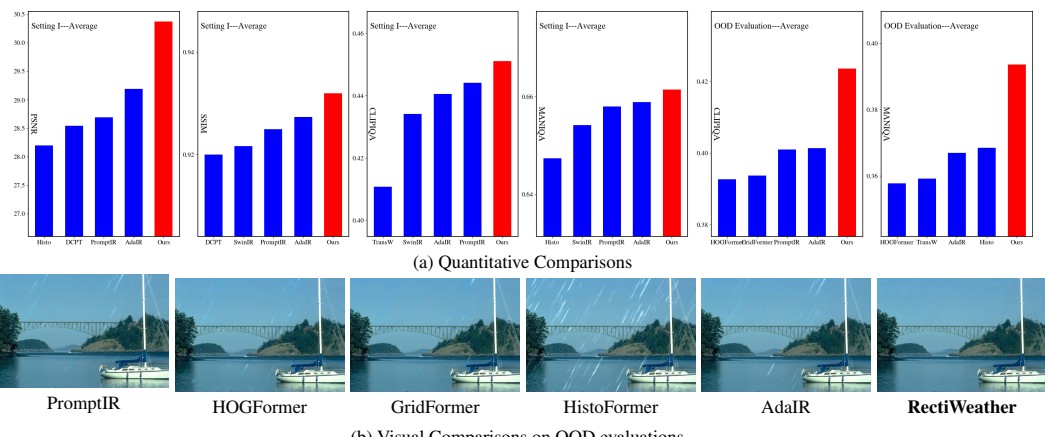

(a) Quantitative Comparisons

PromptIR  HOGFormer  GridFormer  HistoFormer  AdaIR  **RectiWeather**

(b) Visual Comparisons on OOD evaluations.

Figure 1: RectiWeather significantly outperforms SOTA methods, especially on OOD data.

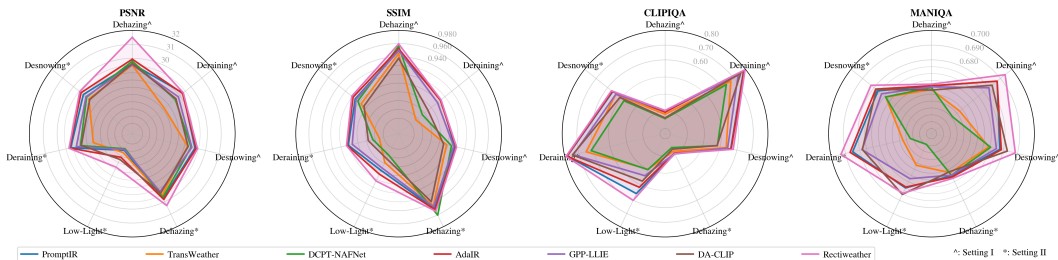

Figure 2: RectiWeather achieves the best performance on both fidelity and perceptual metrics.

iterative denoising process. Despite these advances, two challenges remain, hindering generalization in real-world scenarios and the generation of photo-realistic restorations.

**First**, a key open question is *how to reliably and accurately perceive the underlying weather condition from degraded inputs*. Essentially, such perception plays a pivotal role in the context of all-in-one AWR, as it allows the model to tailor its computational mechanisms or structural adaptations to produce weather-aware outputs with improved performance. To this end, several methods (Potlapalli et al., 2023; Valanarasu et al., 2022) incorporate learnable architectures to implicitly encode weather attributes (*e.g.*, type and severity) for guiding the restoration process, while others opt for explicit weather prediction by training or fine-tuning classifiers (Xu et al., 2024; Hu et al., 2025) or CLIP-based frameworks (Jiang et al., 2024; Luo et al., 2023a; Zeng et al., 2025). While promising within the training distribution, these methods exhibit noticeable performance drops on out-of-distribution (OOD) data, demonstrating their limited generalization ability.

**Second**, distortion-centric training in discriminative frameworks often leads to excessive smoothness, while diffusion-based methods, though perceptually compelling, commonly underperform in fidelity. *Achieving photo-realistic restorations with both minimal distortion and high perceptual quality remains largely unresolved in AWR*.

To address these limitations, we propose **RectiWeather**, an all-in-one AWR framework guided by zero-shot soft weather perceptions derived from pretrained Vision–Language Models (VLMs). **First**, within our developed AWR-specific Question Answering (AWR-QA) module, we craft explicit and unambiguous definitions to refine VLM understanding of weather conditions and low-level attributes, and then quantify their responses into soft perceptions. **Second**, we leverage these perceptions to modulate AWR backbones via Attribute-Modulated Normalization (AMN) and Weather-Weighted Adapters (WWA), enabling degradation-aware posterior estimation under distortion losses. **More importantly**, we introduce a perception-aware rectified flow that approximates the optimal transport map from a perception-dependent source distribution to the clean image distribution, enhancing photo-realism while maintaining competitive fidelity, as demonstrated in Fig. 1.

Our main contributions are: (1) We propose **RectiWeather**, an all-in-one AWR framework guided by **zero-shot** soft perceptions extracted from pretrained VLMs, and equipped with a perception-aware rectified flow to enhance photo-realism. (2) We introduce the AWR-QA module for extracting

Figure 3: The framework of **RectiWeather**. With zero-shot soft perceptions ($\mathcal{P}_{type}$ and $\mathcal{P}_{attr}$) extracted from our developed AWR-QA, we introduce the Attribute-Modulated Normalization (AMN) and Weather-weighted Adapters (WWA) into AWR backbones to enhance fidelity. Subsequently, based on the posterior estimation $\mathbf{x}_0$, we develop a perception-aware rectified flow model $f_\theta$, in which both the source distribution and the velocity field estimation are informed by soft perceptions, enabling photo-realistic restoration.

quantitative weather-aware soft perceptions, and AMN and WWA as **plug-and-play** components for degradation-aware posterior estimation, which improve fidelity and generalize across diverse AWR backbones. (3) We conduct extensive experiments showing that RectiWeather achieves state-of-the-art fidelity and perceptual performance on both in-distribution and out-of-distribution benchmarks.

## 2 METHOD

The key contributions of this work is to introduce zero-shot soft perceptions extracted from VLMs into AWR backbones for fidelity improvement, and to develop a perception-aware rectified flow model to enhance perceptual quality. The overall framework of our method is demonstrated in Fig. 3. Specifically, we first develop the AWR-related Question Answering module to acquire quantified soft perceptions of weather type and low-level visual attributes (Sec. 2.1). Then, we integrate extracted soft perceptions to assist existing AWR baselines via attribute-modulated normalization and weather-weighted adapters (Sec. 2.2). Furthermore, upon high-quality posterior estimation, we introduce a perceptual-aware rectified flow model to achieve the photo-realistic restoration (Sec. 2.3).

### 2.1 AWR-SPECIFIC QUESTION ANSWERING MODULE (AWR-QA)

An all-in-one AWR agent should incorporate dedicated mechanisms for perceiving weather variations in input images, enabling adaptive decisions on model selection (Yang et al., 2024), architecture width (Xu et al., 2024), and computational flow (Potlapalli et al., 2023). Such specially designed modules can generally be divided into two categories: (1) Several approaches (*e.g.*, PromptIR (Potlapalli et al., 2023) and TransWeather Valanarasu et al. (2022)) employ learnable parameters to im-

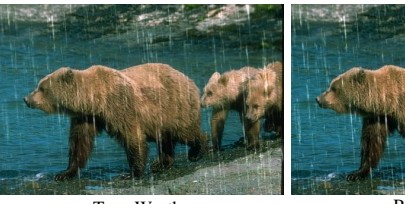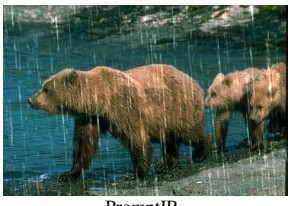

TransWeather   PromptIR

Figure 4: Methods with implicit weather perception present limited weather perception capability on OOD data (R100L (Yang et al., 2017)), leading to poor restoration effect.

plicitly perceive weather type and severity; (2) Some recent methods (*e.g.*, DACLIP (Luo et al., 2023a) and DCPT (Hu et al., 2025)) pursue explicit weather classification through classifier training or CLIP-based fine-tuning, with classification accuracy reported to demonstrate their perception ability. Although effective on in-distribution samples, both implicit learnable modules and fine-tuned classifiers exhibit limited generalization to out-of-distribution (OOD) data, as presented in Fig. 4 and Table. 1. Notably, CLIP-based classifiers, despite originating from VLMs with remarkable zero-shot capacity, show reduced generalization after finetuning (see Table 1), as the adaptation enforces near-perfect classification on comparatively small training datasets.

Accordingly, to fully inherit the zero-shot strength of pretrained VLMs, we directly exploit their perceptual ability to acquire robust perceptions of weather conditions as well as critical visual attributes. Trained on massive image–text corpora, VLMs inherently possess the ability to assess input image quality. Recent studies (Wu et al., 2024; You et al., 2024) have further enhanced their perceptual and interpretive capacity through finetuning on large-scale visual instruction–response

Figure 5: Our developed AWR-specific Question Answering Module.

datasets. While these VLMs exhibit notable success in assessing global image quality, their direct utilization for advancing AWR remains hindered by two critical challenges: (1) Quality perception does not guarantee awareness of weather conditions, thus VLMs demand additional guidance and instructional cues to perform nuanced differentiation of weather-corrupted images; (2) The reliance on one-hot labels for weather perception imposes severe limitations on handling multiple degradations. Yet, prevailing VLMs are trained with one-hot labels, leaving the acquisition of soft perceptions an open challenge. To address these issues, we develop an AWR-specific question answering module as presented in Fig. 5. Specifically, our AWR-QA involves two forms of conversation (type-specific and attribute-specific), and possesses the following distinctive characteristics.

Table 1: The accuracy comparison of weather condition prediction on setting I and OOD data. With our developed AWR-QA module, our pipeline demonstrates markedly improved performance. [Key: †: training or finetuning; "Only Question": Removing Definition from our AWR-QA].

| Methods | Setting I | | | | OOD Data | | | |
|---|---|---|---|---|---|---|---|---|
| | Hazy | Rainy | Snowy | Average | Hazy | Rainy | Snowy | Average |
| Original CLIP | 96.70% | 97.00% | 33.61% | 75.68% | 91.14% | 84.00% | 19.48% | 82.48% |
| Finetuning CLIP† | 99.40% | 99.50% | 99.67% | 99.50% | 92.06% | 82.00% | 90.83% | 91.71% |
| CLIP-Adapter† | 99.80% | 100% | 100% | 99.89% | 81.58% | 91.00% | 91.49% | 82.97% |
| DACLIP† | 100% | 100% | 100% | 100% | 92.48% | 93.00% | 92.31% | 92.47% |
| Only Question | 95.20% | 86.50% | 82.70% | 90.52% | 93.20% | 85.00% | 81.83% | 91.65% |
| AWR-QA | 99.10% | 91.50% | 95.51% | 97.06% | 97.55% | 96.00% | 96.56% | 97.40% |

**Definition-Question-Answering Process** VLM conversations typically adopt the query–response pipeline: `` + `<Question>` → `<Answer>`. However, this pipeline is sub-optimal for AWR task, as evidenced by the low perception accuracy of snowy images in Table 1 ("Only Question"). This observation demonstrates that existing VLMs, while trained to perceive visual quality and degradation, lack explicit grounding in the causal origins of these degradations, making direct weather identification inherently ambiguous. Consequently, the perceptual similarity between snow and haze, both characterized by diminished contrast and visibility, results in frequent misclassification of snowy images as hazy when explicit priors are absent. To address this limitation, we propose to insert textual cues `<Definition>` for the weather condition between `` and `<Question>` to strengthen VLMs' ability to distinguish weather-induced degradations. By injecting `<Definition>` in AWR-QA, we constrain the decision boundary that VLMs otherwise learn only from quality cues, lowering the ambiguity among conditions sharing similar degradations and achieving higher weather condition prediction accuracy in Table 1.

**Producing Soft Perceptions via Multiple Conservations** To facilitate high-quality generalization in real-world scenarios with complex weather conditions (*e.g.*, rain+snow), we aim to produce soft perceptions instead of one-hot label by employing $N$ independent type-specific conversations. In each conversation $i \in [0, N]$, the definition and question for weather type $i$ ( `<Definition>[type[i]]` and `<Question>[type[i]]`) are fed into VLMs and the probability corresponding to weather type $i$ ($\mathcal{P}_{type}^i$) is quantified based on the raw logits of answering tokens. Specifically, the logits for token "yes" and "no" are utilized as anchors for positive and negative responses to the **Yes-or-No** `<Question>[type[i]]`, then the quantified probability $\mathcal{P}_{type}^i$ and the soft weather perception $\mathcal{P}_{type}$ can be calculated as:

$$\mathcal{P}_{type} = \{\mathcal{P}_{type}^i\}, \quad \mathcal{P}_{type}^i = (1 + e^{(L_i^{no} - L_i^{yes})/3})^{-1}, \quad i \in [0, N]. \tag{1}$$

Similarly, besides the type-specific conversations, $M$ attribute-specific dialogues are deployed to perceive low-level attributes for inputs. Specifically, as the the `<Question>[Attr[j]]` is essentially a **How** question, we leverage the logits for token "good" and "poor" as anchors, and calculate

Table 2: Comparison with SOTA methods on all-in-one image restoration (setting I). [Key: Best; Second-best; Third-best].

| Task | Metrics | CNN-based | | Transformer-based | | | | | | | SDE/Diffusion-based | | | Our |
|---|---|---|---|---|---|---|---|---|---|---|---|---|---|---|
| | | WGWS-Net | DCPT-NAFNet | SwinIR | PromptIR | TransWeather | Histoformer | GridFormer | AdaIR | HOGFormer | DACLIP | GPPLLIE | UniRestore | RectiWeather |
| Dehazing | PSNR↑ | 25.2422 | 29.5123 | 28.1335 | 29.0742 | 28.8708 | 28.8753 | 29.7015 | 29.8552 | 26.2312 | 29.1248 | 28.9203 | 24.1892 | 31.8040 |
| | SSIM↑ | 0.9198 | 0.9561 | 0.9490 | 0.9492 | 0.9449 | 0.9532 | 0.9550 | 0.9523 | 0.9446 | 0.9366 | 0.9498 | 0.8272 | 0.9595 |
| | LPIPS↓ | 0.0915 | 0.0245 | 0.0351 | 0.0302 | 0.0327 | 0.0284 | 0.0249 | 0.0290 | 0.0361 | 0.0249 | 0.0336 | 0.1008 | 0.0236 |
| | FID↓ | 13.9170 | 6.6464 | 10.5504 | 8.5657 | 9.6171 | 8.1998 | 5.4800 | 8.2878 | 9.7512 | 5.6563 | 9.3160 | 19.1667 | 4.1463 |
| | MUSIQ↑ | 52.3578 | 56.1662 | 55.9391 | 55.9365 | 56.0669 | 56.1384 | 56.2397 | 55.9506 | 56.0219 | 56.871 | 55.8657 | 54.7502 | 57.1226 |
| | CLIPIQA↑ | 0.2506 | 0.2580 | 0.2944 | 0.3046 | 0.2831 | 0.2591 | 0.2616 | 0.2957 | 0.2557 | 0.2623 | 0.3045 | 0.2530 | 0.3055 |
| | NIQE↓ | 4.3388 | 4.0909 | 3.9766 | 3.9844 | 3.8393 | 4.0926 | 3.9909 | 4.0457 | 4.1116 | 3.8933 | 3.9641 | 4.4329 | 3.6223 |
| | MANIQA↑ | 0.6277 | 0.6398 | 0.6390 | 0.6420 | 0.6367 | 0.6379 | 0.6416 | 0.6423 | 0.6359 | 0.6360 | 0.6408 | 0.6087 | 0.6446 |
| Deraining | PSNR↑ | 25.4435 | 26.6555 | 28.1922 | 28.2102 | 24.3043 | 27.2624 | 26.9545 | 28.2166 | 25.9042 | 26.8019 | 27.3112 | 21.8189 | 28.3725 |
| | SSIM↑ | 0.8057 | 0.8340 | 0.8870 | 0.8881 | 0.8151 | 0.8551 | 0.8397 | 0.8868 | 0.8226 | 0.8504 | 0.8784 | 0.6732 | 0.8886 |
| | LPIPS↓ | 0.1922 | 0.1536 | 0.0748 | 0.0739 | 0.1365 | 0.1331 | 0.1066 | 0.0738 | 0.1626 | 0.0776 | 0.0967 | 0.2254 | 0.0570 |
| | FID↓ | 42.6285 | 54.8526 | 25.3864 | 25.9583 | 50.3034 | 44.4640 | 34.9925 | 25.4323 | 57.4706 | 29.3110 | 30.4662 | 67.9509 | 21.6385 |
| | MUSIQ↑ | 63.9255 | 64.9781 | 70.4030 | 70.7116 | 68.8380 | 67.7278 | 68.7697 | 70.5658 | 66.0232 | 70.3025 | 70.2249 | 64.6621 | 70.8914 |
| | CLIPIQA↑ | 0.6303 | 0.6326 | 0.7695 | 0.7827 | 0.6721 | 0.6856 | 0.7025 | 0.7871 | 0.6623 | 0.7589 | 0.7359 | 0.6585 | 0.7962 |
| | NIQE↓ | 3.5515 | 3.5908 | 3.6381 | 3.6483 | 3.7658 | 3.4542 | 3.4429 | 3.5867 | 3.3721 | 3.1197 | 3.5928 | 4.5751 | 3.1935 |
| | MANIQA↑ | 0.6208 | 0.6119 | 0.6859 | 0.6871 | 0.6253 | 0.6417 | 0.6383 | 0.6871 | 0.6149 | 0.6827 | 0.6798 | 0.6135 | 0.6941 |
| Desnowing | PSNR↑ | 26.2841 | 27.5454 | 27.4389 | 28.2108 | 26.9457 | 27.3890 | 27.5816 | 28.4096 | 26.2566 | 27.0387 | 27.6699 | 21.8825 | 28.6361 |
| | SSIM↑ | 0.8330 | 0.8886 | 0.8874 | 0.8966 | 0.8769 | 0.8870 | 0.8794 | 0.8992 | 0.8794 | 0.8701 | 0.8938 | 0.6402 | 0.9005 |
| | LPIPS↓ | 0.1145 | 0.0808 | 0.0705 | 0.0627 | 0.0790 | 0.0846 | 0.0734 | 0.0590 | 0.0886 | 0.0677 | 0.0668 | 0.1145 | 0.0537 |
| | FID↓ | 27.2584 | 29.1007 | 25.1864 | 22.1090 | 29.7221 | 30.7080 | 23.6043 | 20.8006 | 32.8779 | 23.4294 | 25.3291 | 40.0907 | 17.2547 |
| | MUSIQ↑ | 69.0165 | 69.7187 | 68.6227 | 69.1966 | 69.1384 | 70.1404 | 69.1025 | 69.2825 | 69.0672 | 70.6822 | 69.4237 | 68.5674 | 70.4982 |
| | CLIPIQA↑ | 0.4787 | 0.4836 | 0.5552 | 0.5638 | 0.5362 | 0.4947 | 0.5017 | 0.5663 | 0.4864 | 0.4819 | 0.5452 | 0.4484 | 0.5784 |
| | NIQE↓ | 3.1464 | 3.0592 | 2.8550 | 2.9555 | 2.9998 | 3.1293 | 2.8889 | 2.9380 | 3.0156 | 2.9284 | 3.0459 | 3.3874 | 2.6762 |
| | MANIQA↑ | 0.6645 | 0.6611 | 0.6688 | 0.6749 | 0.6584 | 0.6651 | 0.6681 | 0.6771 | 0.6601 | 0.6829 | 0.6698 | 0.6356 | 0.6885 |
| Average | PSNR↑ | 25.6122 | 28.5387 | 27.9082 | 28.6901 | 27.7213 | 28.2002 | 28.6890 | 29.1908 | 26.2034 | 28.1707 | 28.3243 | 23.1562 | 30.3658 |
| | SSIM↑ | 0.8782 | 0.9200 | 0.9216 | 0.9249 | 0.9078 | 0.9202 | 0.9202 | 0.9273 | 0.9093 | 0.9048 | 0.9232 | 0.7477 | 0.9319 |
| | LPIPS↓ | 0.1104 | 0.0576 | 0.0513 | 0.0459 | 0.0597 | 0.0588 | 0.0502 | 0.0440 | 0.0677 | 0.0450 | 0.0517 | 0.1192 | 0.0374 |
| | FID↓ | 21.5575 | 19.4927 | 17.0820 | 15.0166 | 20.8444 | 19.7380 | 14.8055 | 14.3673 | 22.7679 | 14.2141 | 17.0083 | 31.5666 | 10.4631 |
| | MUSIQ↑ | 59.2015 | 61.6673 | 61.7779 | 62.0022 | 61.8471 | 62.0979 | 61.9235 | 62.0225 | 61.4858 | 62.9714 | 61.9846 | 60.4618 | 63.1151 |
| | CLIPIQA↑ | 0.3689 | 0.3749 | 0.4342 | 0.4442 | 0.4108 | 0.3851 | 0.3907 | 0.4406 | 0.3778 | 0.3907 | 0.4327 | 0.3632 | 0.4511 |
| | NIQE↓ | 3.8535 | 3.6911 | 3.5647 | 3.6037 | 3.5510 | 3.7002 | 3.5623 | 3.6251 | 3.6637 | 3.4854 | 3.6165 | 4.0998 | 3.2590 |
| | MANIQA↑ | 0.6392 | 0.6438 | 0.6542 | 0.6580 | 0.6427 | 0.6474 | 0.6501 | 0.6589 | 0.6416 | 0.6568 | 0.6548 | 0.6182 | 0.6647 |

the attribute perception $\mathcal{P}_{attr}$ as:

$$\mathcal{P}_{attr} = \{\mathcal{P}_{attr}^{j}\}, \quad \mathcal{P}_{attr}^{j} = (1 + e^{(L_i^{poor} - L_i^{good})/3})^{-1}, \quad j \in [0, M]. \tag{2}$$

Therefore, with our soft weather and attribute perceptions, $\mathcal{P}_{type}$ tends to produce a non-sharp distribution under unseen weather conditions, facilitating expert mixing behavior, while $\mathcal{P}_{attr}$ directly encodes low-level degradation intensity, alleviating error propagation from weather misclassification.

**Remarkable Zero-shot Capability of AWR-QA**    To evaluate the effectiveness of our developed AWR-QA, we convert soft weather perceptions into one-hot labels and compare the statistical accuracy against existing methods in Table 1. Though training or finetuning CLIP-based approaches achieves slightly higher performance, their performance drop heavily on OOD data while our AWR-QA is capable of achieve consistent perceptions across various weather conditions.

## 2.2 ENHANCING AWR BASELINES USING WEATHER PERCEPTIONS

With robust weather and visual attribute perceptions extracted from VLMs, we aim to enhance various baseline models using $\mathcal{P}_{type}$ and $\mathcal{P}_{attr}$ in this section. As presented in Fig. 3, we take transformer-based baselines as the example for illustration. Overall, two additional layers are introduced into the Soft Perceptions guided Transformer Block (SP-TransB).

**Attributed-Modulated Normalization (AMN)**    The attribute perception is integrated to modulate the layer normalization. Specifically, given an input feature $\mathbf{F}$ or $\mathbf{F}_{mid}$, AMN modulate its output of standard layer normalization with the scale and bias, which are learned from $\mathcal{P}_{attr}$ via linear operation. This design allows the adaptive modulate magnitude based on low-level attribute perception and alleviates the error propagation when the weather perception is not accurate in extremely complex scenarios. The calculation process of AMN can be represented as:

$$\lambda_1, \beta_1, \lambda_2, \beta_2 = \text{Linear}(\mathcal{P}_{attr}), \tag{3}$$

$$\tilde{\mathbf{F}} = \lambda_1 \odot \text{LN}(\mathbf{F}) + \beta_1, \quad \tilde{\mathbf{F}}_{mid} = \lambda_2 \odot \text{LN}(\mathbf{F}_{mid}) + \beta_2. \tag{4}$$

**Weather-Weighted Adapters (WW-Adapter)**
To achieve weather-aware modeling, we incorporate soft weather perceptions $\mathcal{P}_{type}$ by developing the WW-Adapter alongside the MSA and FFN components. The details of WW-Adapter is shown in Fig. 6. Specifically, the WW-Adapter

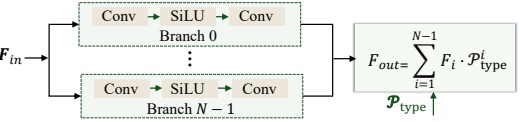

Figure 6: Soft weather perceptions $\mathcal{P}_{type}$ serve as branch weight in our developed WW-Adapter.

module contains $N$ branches, with each corresponding to one weather type. The output of WW-Adapter is calculated as the weighted sum of the output of each branch:

$$\mathbf{F}_{out} = \sum_{i=0}^{N-1} \mathbf{F}_i \cdot \mathcal{P}_{\text{type}}^i, \quad \mathbf{F}_i = [\text{Conv}, \text{SiLU}, \text{Conv}]^i(\mathbf{F}_{in}). \quad (5)$$

Therefore, the calculation of SP-TransB can be formulated as:

$$\tilde{\mathbf{F}} = \text{AMN}(\mathbf{F}), \quad \mathbf{F}_A^{MSA} = \text{WWA}(\tilde{\mathbf{F}}, \mathcal{P}_{type}), \quad \mathbf{F}_m = \mathbf{F} + \text{MSA}(\tilde{\mathbf{F}}) + \alpha_A^{MSA} \cdot \mathbf{F}_A^{MSA}, \quad (6)$$

$$\tilde{\mathbf{F}}_m = \text{AMN}(\mathbf{F}_m), \mathbf{F}_A^{FFN} = \text{WWA}(\tilde{\mathbf{F}}_m, \mathcal{P}_{type}), \quad \mathbf{F}' = \mathbf{F}_m + \text{FFN}(\tilde{\mathbf{F}}_m) + \alpha_A^{FFN} \cdot \mathbf{F}_A^{FFN}, \quad (7)$$

where $\alpha_A^{MSA}$ and $\alpha_A^{FFN}$ are two learnable coefficients for adaptively adjusting the strength of weather-weighted adapters. Following TransWeather (Valanarasu et al., 2022), the optimization of AWR backbones with SP-TransB is achieved by minimizing the following distortion loss $\mathcal{L}_{dist}$:

$$\mathcal{L}_{dist} = \mathcal{L}_1(\mathbf{x}_1, \mathbf{x}_0) + 0.04 \times \mathcal{L}_{percep}(\mathbf{x}_1, \mathbf{x}_0), \quad (8)$$

where $\mathbf{x}_1$ and $\mathbf{x}_0$ denote the clean image and posterior output of AWR backbones with SP-TransB.

## 2.3 WEATHER-AWARE RECTIFIED FLOW

Using soft weather cues as conditioning, attribute-modulated normalization together with weather-weighted adapters consistently improves fidelity metrics over the corresponding AWR backbones. Nevertheless, the posterior estimate $\mathbf{x}_0$ exhibits over-smoothing and lacks photorealism. We therefore learn a weather-aware rectified flow to map the posterior-estimate distribution to the clean-image distribution, which differs from the original RF process (Liu et al., 2022) in the following aspects.

**Parameterizing Source Distribution using Soft Perceptions**   RF approximates a straight-path flow that effects a distributional transport from source to target. In our setting, the target distribution is the clean-image distribution, from which clean images $\mathbf{x}_1$ are sampled. However, the choice of source distribution and its sampling scheme merits careful investigation. The standard RF, trained from Gaussian noise to natural images, is effective for generation yet underperforms in AWR owing to limited data-consistency. A similar limitation arises in approaches that take the degraded observation $\mathbf{y}$ as the source for flow learning (Albergo & Vanden-Eijnden, 2022). PMRF (Ohayon et al., 2024) defines the source by perturbing posterior estimates with random noise and demonstrates efficacy on face restoration, denoising, and super-resolution. This stochastic perturbation mitigates singularities inherent to learning a strictly deterministic mapping between source and target. Given the adverse impact of noise on fidelity, PMRF employs modest, task-dependent noise magnitudes, indicating that a constant noise level is ill-suited to AWR. To this end, we propose to parameterize the source distribution exclusively through soft perceptions extracted by our AWR-QA.

Concretely, we first compute a normalized entropy $H$ for weather perceptions $\mathcal{P}_{type} \in \mathbb{R}^N$, and aggregate attribute perceptions $\mathcal{P}_{attr} \in \mathbb{R}^M$ into a scalar degradation severity indicator. We then We then fuse type-uncertainty and attribute-severity and calculate a scalar perturbation scale within a prescribed range $[\delta_{min}, \delta_{max}]$. The overall process can be summarized as:

$$\tilde{\mathcal{P}}_{type} = \frac{\mathcal{P}_{type}}{\sum_{i=0}^{N-1} \mathcal{P}_{type}^i}, \quad H = -\frac{\sum_{i=0}^{N-1} \mathcal{P}_{type}^i \log \mathcal{P}_{type}^i}{\log N}, \quad s_{attr} = 1 - \frac{\sum_{i=0}^{M-1} \mathcal{P}_{attr}^i}{M}, \quad (9)$$

$$u = \alpha H + (1 - \alpha) s_{attr}, \quad \delta = \delta_{min} + (\delta_{max} - \delta_{min}) u, \quad (10)$$

where $\alpha$ is set to 0.5 to control the trade-off between weather uncertainty and attribute severity. $\delta_{min}$ and $\delta_{max}$ are 0.025 and 0.1, respectively. Therefore, as presented in Eq. 11, the source distribution in our weather-aware rectified flow is isotropic Gaussian centered at posterior estimate $\mathbf{x}_o$ and parameterized by a VLM-driven perceptions, ensuring data-consistent yet adaptive randomness.

$$p_0^* = \mathcal{N}(\mathbf{x}_0, \delta^2 I), \quad \mathbf{x}_0^* = \mathbf{x}_0 + \delta z, \quad z \sim \mathcal{N}(0, I). \quad (11)$$

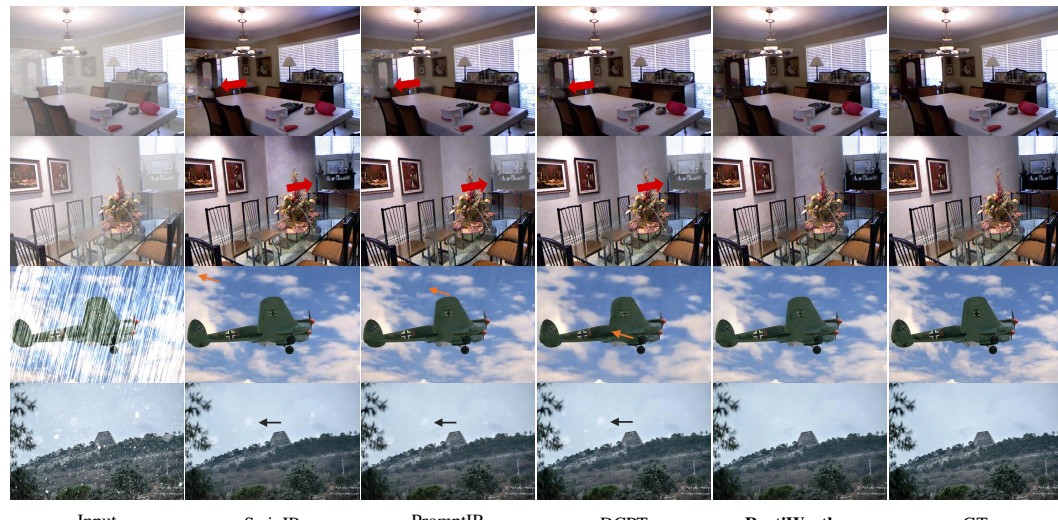

| Input | SwinIR | PromptIR | DCPT | **RectiWeather** | GT |

Figure 7: Visual comparisons on setting I. Our RectiWeather is capable of preserving fine details and maintaining perceptual quality.

Table 3: Our extracted soft degradation perceptions consistently achieves remarkable improvements for CNN or Transformer networks on setting I.

| Methods | Dehazing | | Deraining | | Desnowing | | Average | |
|---|---|---|---|---|---|---|---|---|
| | PSNR ↑ | SSIM ↑ | PSNR ↑ | SSIM ↑ | PSNR ↑ | SSIM ↑ | PSNR ↑ | SSIM ↑ |
| NAFNet | 29.2794 | 0.9481 | 26.0314 | 0.8494 | 28.1914 | 0.8953 | 28.5554 | 0.9195 |
| **NAFNet+Prior (Ours)** | 31.3548 | 0.9572 | 26.5934 | 0.8617 | 28.6328 | 0.8982 | 30.0240 | 0.9269 |
| TransWeather | 28.8708 | 0.9449 | 24.3043 | 0.8151 | 26.9457 | 0.8769 | 27.7213 | 0.9078 |
| **TransWeather+Prior (Ours)** | 30.7832 | 0.9574 | 26.3847 | 0.8557 | 28.1937 | 0.8943 | 29.4306 | 0.9251 |
| SwinIR | 28.133 | 0.9490 | 28.1922 | 0.8870 | 27.4389 | 0.8873 | 27.9082 | 0.9216 |
| **SwinIR+Prior (Ours)** | 29.2637 | 0.9542 | 28.3142 | 0.8881 | 27.8045 | 0.8891 | 28.6713 | 0.9251 |
| PromptIR | 29.0742 | 0.9492 | 28.2102 | 0.8881 | 28.2108 | 0.8966 | 28.6901 | 0.9249 |
| **PromptIR+Prior (Ours)** | 30.1556 | 0.9530 | 27.9903 | 0.8863 | 28.6774 | 0.9034 | 29.4219 | 0.9291 |
| Histoformer | 28.8753 | 0.9532 | 27.2624 | 0.8551 | 27.3890 | 0.8870 | 28.2002 | 0.9202 |
| **Histoformer+Prior (Ours)** | 29.8024 | 0.9625 | 27.6263 | 0.8612 | 27.9804 | 0.8938 | 28.9527 | 0.9283 |
| AdaIR | 29.8552 | 0.9523 | 28.2166 | 0.8868 | 28.4096 | 0.8992 | 29.1908 | 0.9273 |
| **AdaIR+Prior (Ours)** | 31.9523 | 0.9614 | 28.5753 | 0.8892 | 28.7732 | 0.9009 | 30.5164 | 0.9332 |

**Perception-aware Velocity Field Estimation**     Based on the source distribution $p_0^*$ and its samples $\mathbf{x}_0^*$, we train a perception-aware rectified flow model $f_\theta$ via the following optimization:

$$\arg\min_\theta \int_0^1 \mathbb{E}\left[||\,(\mathbf{x}_1 - \mathbf{x}_0^*) - f_\theta(\mathbf{x}_t^*, t, \mathcal{P}_{type}, \mathcal{P}_{attr})||^2\right] dt, \tag{12}$$

where $t$ is sampled from a uniform distribution $\mathcal{U}(0,1)$, $\mathbf{x}_t^* = t\mathbf{x}_1 + (1-t)\mathbf{x}_0^*$. More importantly, other than learning a unified velocity path for all weather-corrupted inputs, we incorporate soft perceptions into $f_\theta$ to achieve perception-aware velocity field estimation. Overall, AdaIR (Cui et al., 2025) is deployed as the backbone of $f_\theta$, weather perceptions $\mathcal{P}_{type}$ is introduced via weather-weighted adapters. Moreover, the normalization coefficients of AMN in Eq. 4 is calculated upon the concatenation of timestep $t$ and weather perceptions $\mathcal{P}_{attr}$.

## 3 EXPERIMENTS

**Experiment Settings and Datasets**     We evaluate RectiWeather under two all-in-one experimental settings. Setting I involves three core weather degradation tasks, dehazing, deraining, and desnowing, trained on a unified dataset comprising Reside-6K (Li et al., 2018), Rain100H (Yang et al., 2017), and Snow100K-L(Liu et al., 2018). Setting II further introduces the low-light enhancement task, with LOLv2-Real (Yang et al., 2021) included as an additional domain. To assess cross-domain robustness,

Table 4: Comparison with OOD data using weights pre-trained on setting I. [Key: Best; Second-best; Third-best].

| Tasks | Metrics | Baselines | | | | | | Ours |
|---|---|---|---|---|---|---|---|---|
| | | PromptIR | TransWeather | Histoformer | GridFormer | AdaIR | HOGFormer | RectiWeather |
| *Dehazing* | MUSIQ↑ | 51.4000 | 51.9740 | 52.7529 | 54.6239 | 54.5125 | 52.8634 | 57.2632 |
| | CLIPIQA↑ | 0.3677 | 0.3615 | 0.3642 | 0.3659 | 0.3677 | 0.3594 | 0.3921 |
| | NIQE↓ | 5.3796 | 5.1483 | 4.8733 | 4.9918 | 4.9563 | 5.0271 | 4.4625 |
| | MANIQA↑ | 0.2893 | 0.3079 | 0.3164 | 0.3209 | 0.3135 | 0.3029 | 0.3426 |
| *Deraining* | MUSIQ↑ | 71.0463 | 70.4322 | 70.6811 | 70.6135 | 71.2161 | 70.2886 | 71.6847 |
| | CLIPIQA↑ | 0.7526 | 0.7242 | 0.7156 | 0.7388 | 0.7570 | 0.7229 | 0.7689 |
| | NIQE↓ | 3.3537 | 3.4932 | 3.8112 | 3.5161 | 3.3245 | 3.4784 | 2.9683 |
| | MANIQA↑ | 0.6978 | 0.6628 | 0.6868 | 0.6866 | 0.6984 | 0.6852 | 0.7065 |
| *Desnowing* | MUSIQ↑ | 70.4422 | 70.3448 | 71.1404 | 70.1392 | 70.4569 | 70.5921 | 71.3847 |
| | CLIPIQA↑ | 0.5793 | 0.5227 | 0.5333 | 0.5336 | 0.5815 | 0.5742 | 0.5898 |
| | NIQE↓ | 2.9182 | 3.1424 | 2.9812 | 3.0258 | 2.9156 | 2.9046 | 2.8125 |
| | MANIQA↑ | 0.6897 | 0.6724 | 0.6850 | 0.6749 | 0.6915 | 0.6923 | 0.7033 |
| *Average* | MUSIQ↑ | 54.1020 | 54.5709 | 55.3413 | 56.8251 | 56.7800 | 55.3619 | 59.2641 |
| | CLIPIQA↑ | 0.4010 | 0.3883 | 0.3917 | 0.3937 | 0.4014 | 0.3927 | 0.4236 |
| | NIQE↓ | 5.0405 | 4.8719 | 4.6225 | 4.7238 | 4.6761 | 4.7387 | 4.2325 |
| | MANIQA↑ | 0.3460 | 0.3592 | 0.3685 | 0.3711 | 0.3670 | 0.3578 | 0.3936 |

we conduct out-of-distribution evaluations by testing models trained in Setting I on RTTS (Li et al., 2018), Rain100L(Yang et al., 2017), and Snow100K-S(Liu et al., 2018), each representing unseen test conditions for dehazing, deraining, and desnowing respectively.

**Implementations Details** Using Eq. 8, we first train the AWR baseline model (AdaIR (Cui et al., 2025)) with AMN and WWA on our combined dataset for 200 epochs. The initial learning rate is set to $2 \times 10^{-4}$ and it is reduced to $10^{-7}$ by the end of training. Each training input is cropped into $192 \times 192$, and the batch size is set to 4. We use horizontal flips and rotations for data augmentation. We then optimize $f_\theta$ using Eq. 12 for 500 epochs, while other configurations keep unchanged. As in (Ohayon et al., 2024), we sample $t$ uniformly from $U[0, 1]$ using a stratified sampling strategy. In the inference stage, the number of flow steps $K$ is set to 5 in this work.

**Compared Baselines and Metrics** We benchmark our method against state-of-the-art (SOTA) approaches in all-in-one image restoration tasks including CNN-based methods (NAFNet (Chen et al., 2022), WGWS-Net (Zhu et al., 2023), DCPT-NAFNet (Hu et al., 2025)), Transformer-based methods (SwinIR (Liang et al., 2021), PromptIR (Potlapalli et al., 2023), TransWeather (Valanarasu et al., 2022), Histoformer (Sun et al., 2024), GridFormer (Wang et al., 2024), AdaIR (Cui et al., 2025), HOGFormer ()), and SDE/Diffusion-based methods (DACLIP (Luo et al., 2023a), GPPLLIE (Zhou et al., 2025), and UniRestore (Chen et al., 2025)). To test the efficacy of our method against these approaches, we compare performance across dehazing, deraining, desnowing, and low-light conditions across in and out-of-distribution settings, using fidelity (PSNR, SSIM (Wang et al., 2004)) and perceptual metrics (LPIPS (Zhang et al., 2018), FID (Heusel et al., 2017), MUSIQ (Ke et al., 2021), CLIPIQA (Wang et al., 2023), NIQE (Mittal et al., 2012), MANIQA (Yang et al., 2022)).

**Quantitative Comparisons with Baselines** Tab. 2 summarizes the quantitative comparisons between our method and current SOTA methods on setting I. Our method achieves superior performance on all three tasks, highlighting its advantage. Notably, the PSNR of our method surpasses the best SOTA by 1.17 dB on average. Moreover, our perceptual metrics significantly outperform the CNN-based and Transformer-based baselines, with only a few metrics slightly below DACLIP, a generative approach that requires numerous inference steps. These fidelity results demonstrate the effectiveness of the soft perceptions extracted from VLMs and of our designed AMN and WWA modules, while the exceptional perceptual quality corroborates the importance of our perception-aware rectified-flow model. See Tab. 5 for quantitative comparisons on Setting II, where RectiWeather also achieves the best performance on both fidelity and perceptual metrics. To demonstrate the plug-and-play nature of our perceptions, we integrate them into diverse baselines and achieve consistent gains on weather removal tasks (Tab. 3).

**Visual Comparisons with Bselines** Fig. 7 provides visual comparisons between our method and AWR baselines. For the hazy scenes (rows 1–2), SwinIR, PromptIR, and DCPT leave residual haze

Table 5: Comparison with SOTA methods on all-in-one image restoration (setting II). [Key: Best; Second-best; Third-best].

| Tasks | Metrics | CNN-based | | Transformer-based | | | | | | | SDE/Diffusion-based | | | Our |
|---|---|---|---|---|---|---|---|---|---|---|---|---|---|---|
| | | NAFNet | DCPT-NAFNet | SwinIR | PromptIR | TransWeather | Histoformer | GridFormer | AdaIR | HOGFormer | DACLIP | GPP-LLIE | UniRestore | RectiWeather |
| Dehazing | PSNR↑ | 29.4842 | 29.2242 | 26.3268 | 29.3953 | 28.8925 | 27.2265 | 27.0659 | 29.6063 | 27.0072 | 28.4270 | 28.1868 | 24.1414 | 30.3797 |
| | SSIM↑ | 0.9509 | 0.9604 | 0.9389 | 0.9497 | 0.9448 | 0.9418 | 0.9535 | 0.9512 | 0.9493 | 0.9365 | 0.9462 | 0.8346 | 0.9539 |
| | LPIPS↓ | 0.0291 | 0.0246 | 0.0444 | 0.0294 | 0.0330 | 0.0339 | 0.0260 | 0.0296 | 0.0337 | 0.0272 | 0.0368 | 0.0952 | 0.0253 |
| | FID↓ | 8.1313 | 6.9793 | 13.8706 | 8.4712 | 9.5456 | 10.1239 | 6.8404 | 8.3922 | 9.0061 | 6.2342 | 9.9210 | 18.5698 | 5.5366 |
| | MUSIQ↑ | 55.9530 | 56.1014 | 56.3110 | 55.8436 | 55.9373 | 55.6795 | 56.1424 | 55.9427 | 55.8376 | 56.8943 | 55.6146 | 56.3163 | 56.7342 |
| | CLIPIQA↑ | 0.2853 | 0.2598 | 0.2939 | 0.3002 | 0.2832 | 0.2561 | 0.2630 | 0.2950 | 0.2604 | 0.2700 | 0.2987 | 0.2659 | 0.2986 |
| | NIQE↓ | 4.0587 | 4.0638 | 3.8112 | 3.9647 | 3.8340 | 4.0335 | 3.9798 | 4.0271 | 4.1015 | 3.8247 | 3.9758 | 4.4224 | 3.9401 |
| | MANIQA↑ | 0.6416 | 0.6396 | 0.6386 | 0.6403 | 0.6356 | 0.6322 | 0.6392 | 0.6424 | 0.6415 | 0.6338 | 0.6394 | 0.6213 | 0.6445 |
| Low-light | PSNR↑ | 18.6269 | 20.2516 | 16.1077 | 20.5389 | 20.9644 | 17.4699 | 19.9056 | 21.8332 | 19.9568 | 22.2302 | 20.3946 | 19.6627 | 23.6634 |
| | SSIM↑ | 0.8383 | 0.8183 | 0.7913 | 0.8575 | 0.8377 | 0.7956 | 0.8112 | 0.8669 | 0.8318 | 0.8279 | 0.8517 | 0.7387 | 0.8842 |
| | LPIPS↓ | 0.1153 | 0.1887 | 0.1384 | 0.0867 | 0.1176 | 0.1995 | 0.1249 | 0.0849 | 0.1676 | 0.1151 | 0.1157 | 0.1714 | 0.0644 |
| | FID↓ | 43.3469 | 68.5468 | 62.8596 | 33.1933 | 46.6892 | 73.0903 | 49.2181 | 34.4985 | 57.8767 | 47.1134 | 43.2211 | 71.6247 | 29.3878 |
| | MUSIQ↑ | 70.4247 | 55.9111 | 69.2740 | 68.9156 | 66.8728 | 58.9973 | 65.7178 | 68.9272 | 60.1414 | 69.1371 | 68.6057 | 66.7391 | 69.5077 |
| | CLIPIQA↑ | 0.5366 | 0.4023 | 0.5277 | 0.5571 | 0.4001 | 0.4297 | 0.4743 | 0.5162 | 0.3446 | 0.4763 | 0.4415 | 0.4795 | 0.6010 |
| | NIQE↓ | 4.3401 | 3.9823 | 4.3985 | 4.5524 | 3.9808 | 3.8475 | 4.4756 | 4.5544 | 3.8952 | 4.9906 | 3.8835 | 4.6244 | 3.9343 |
| | MANIQA↑ | 0.6575 | 0.5897 | 0.6358 | 0.6601 | 0.6236 | 0.5912 | 0.6210 | 0.6587 | 0.5984 | 0.6706 | 0.6452 | 0.6328 | 0.6687 |
| Deraining | PSNR↑ | 25.4421 | 26.3477 | 26.7710 | 27.9980 | 24.1490 | 25.5034 | 25.7722 | 28.1874 | 26.3770 | 26.1871 | 27.0394 | 21.6014 | 28.3006 |
| | SSIM↑ | 0.8456 | 0.8253 | 0.8580 | 0.8829 | 0.8101 | 0.8069 | 0.8063 | 0.8857 | 0.8346 | 0.8360 | 0.8737 | 0.6743 | 0.8871 |
| | LPIPS↓ | 0.1126 | 0.1648 | 0.1064 | 0.0755 | 0.1428 | 0.1881 | 0.1470 | 0.0733 | 0.1560 | 0.0876 | 0.1017 | 0.2255 | 0.0647 |
| | FID↓ | 39.5993 | 58.2514 | 36.7157 | 27.1002 | 51.7666 | 62.6183 | 45.6402 | 25.8660 | 55.5853 | 32.7593 | 31.7345 | 69.8020 | 19.4836 |
| | MUSIQ↑ | 70.0800 | 64.5522 | 69.4326 | 70.4612 | 68.4701 | 65.4731 | 66.8804 | 70.6815 | 65.9262 | 69.9443 | 70.0148 | 65.8471 | 71.1827 |
| | CLIPIQA↑ | 0.7535 | 0.6158 | 0.7252 | 0.7829 | 0.6509 | 0.6393 | 0.6726 | 0.7856 | 0.6749 | 0.7466 | 0.7402 | 0.6629 | 0.7915 |
| | NIQE↓ | 3.6333 | 3.6589 | 3.6001 | 3.5691 | 3.7411 | 3.2971 | 3.5996 | 3.6318 | 3.3972 | 3.1016 | 3.6051 | 3.6589 | 3.2237 |
| | MANIQA↑ | 0.6707 | 0.6042 | 0.6650 | 0.6850 | 0.6153 | 0.5982 | 0.6150 | 0.6868 | 0.6191 | 0.6761 | 0.6767 | 0.6199 | 0.6932 |
| Desnowing | PSNR↑ | 28.2893 | 27.2178 | 26.1904 | 27.9105 | 26.7268 | 26.5015 | 26.4140 | 28.4000 | 26.7946 | 26.5609 | 27.3473 | 21.8470 | 28.6734 |
| | SSIM↑ | 0.8966 | 0.8843 | 0.8697 | 0.8919 | 0.8743 | 0.8698 | 0.8739 | 0.8981 | 0.8801 | 0.8661 | 0.8895 | 0.6415 | 0.9019 |
| | LPIPS↓ | 0.0625 | 0.0857 | 0.0935 | 0.0665 | 0.0813 | 0.1057 | 0.0945 | 0.0608 | 0.0897 | 0.0731 | 0.0705 | 0.1819 | 0.0527 |
| | FID↓ | 21.4393 | 30.7876 | 34.3009 | 22.9940 | 30.7250 | 36.8774 | 29.5384 | 21.1764 | 33.3173 | 25.8008 | 26.4275 | 45.5577 | 18.0723 |
| | MUSIQ↑ | 69.5727 | 69.5594 | 67.9504 | 68.8747 | 69.1820 | 69.3072 | 68.2443 | 69.1902 | 69.1780 | 70.3695 | 69.0375 | 68.8181 | 70.6482 |
| | CLIPIQA↑ | 0.5652 | 0.4776 | 0.5423 | 0.5621 | 0.5379 | 0.4762 | 0.4991 | 0.5655 | 0.4839 | 0.5082 | 0.5383 | 0.4556 | 0.5711 |
| | NIQE↓ | 2.9698 | 3.0606 | 2.9673 | 2.8967 | 2.9637 | 3.2416 | 2.9067 | 2.9322 | 3.0508 | 2.8812 | 3.0507 | 3.0606 | 2.8239 |
| | MANIQA↑ | 0.6760 | 0.6583 | 0.6568 | 0.6726 | 0.6580 | 0.6568 | 0.6595 | 0.6767 | 0.6581 | 0.6758 | 0.6658 | 0.6382 | 0.6826 |
| Average | PSNR↑ | 28.1100 | 27.8153 | 25.7928 | 28.3130 | 27.2917 | 26.3028 | 26.3470 | 28.6668 | 26.5028 | 27.2754 | 27.3908 | 22.9132 | 29.2682 |
| | SSIM↑ | 0.9167 | 0.9147 | 0.9007 | 0.9195 | 0.9027 | 0.9054 | 0.9054 | 0.9231 | 0.9092 | 0.8980 | 0.9157 | 0.7516 | 0.9268 |
| | LPIPS↓ | 0.0530 | 0.0673 | 0.0714 | 0.0490 | 0.0643 | 0.0815 | 0.0656 | 0.0470 | 0.0713 | 0.0527 | 0.0584 | 0.1403 | 0.0402 |
| | FID↓ | 17.5018 | 23.1392 | 25.3101 | 16.3230 | 22.6373 | 27.4171 | 24.1633 | 15.6456 | 26.3276 | 17.3612 | 19.1862 | 35.2829 | 12.2218 |
| | MUSIQ↑ | 62.5064 | 61.2352 | 62.0532 | 62.1889 | 62.0184 | 61.1928 | 61.3430 | 62.3646 | 61.1712 | 63.1715 | 62.0566 | 61.8197 | 63.3251 |
| | CLIPIQA↑ | 0.4363 | 0.3736 | 0.4301 | 0.4473 | 0.4086 | 0.3751 | 0.3919 | 0.4438 | 0.3791 | 0.4063 | 0.4284 | 0.3789 | 0.4525 |
| | NIQE↓ | 3.6845 | 3.6998 | 3.5531 | 3.6163 | 3.5568 | 3.6959 | 3.6266 | 3.6671 | 3.6844 | 3.5117 | 3.6395 | 3.9222 | 3.5115 |
| | MANIQA↑ | 0.6564 | 0.6392 | 0.6470 | 0.6563 | 0.6399 | 0.6342 | 0.6421 | 0.6588 | 0.6429 | 0.6535 | 0.6520 | 0.6271 | 0.6629 |

or introduce color distortion, as indicated by the red arrows. In contrast, our RectiWeather produces clearer and more natural results, closely resembling the ground truth. For the deraining case (row 3), competing methods fail to fully remove rain streaks or generate noticeable artifacts (orange arrows), whereas RectiWeather removes rain effectively while preserving fine structural details. In the desnowing case (row 4), other baselines fail to eliminate snowy particles and tend to blur background structures (black arrows). By comparison, RectiWeather yields sharper contours and more faithful textures, demonstrating its robustness across diverse weather degradations.

## 4 CONCLUSIONS

In this work, We introduce RectiWeather for adverse weather removal. By leveraging VLM-guided soft perceptions for restoration models, RectiWeather enhances the AWR baselines' awareness and adaptability to complex weather degradations without requiring additional supervision. Furthermore, with improved posterior estimation through perception-aware rectified flow, our method achieves significant performance in both all-in-one and out-of-distribution scenarios. Extensive experiments demonstrate RectiWeather's state-of-the-art performance in fidelity and perceptual quality.

**Limitations** Although RectiWeather handles complex weather conditions better than baseline models, its performance remains limited when image degradation (*e.g.*, the haze pattern) differs substantially from those seen during training. In addition, while our method is more efficient than full diffusion-based approaches such as DACLIP, the incorporation of VLM perception in the adverse-weather removal (AWR) backbone and rectified-flow procedure introduce a modest runtime overhead; for a 1024×1024 image, processing typically requires 0.4 s of additional time on a single 5090 GPU.

## 5 REPRODUCIBILITY STATEMENT

We have made significant efforts to ensure the reproducibility of our results. The main text provides complete details of the proposed framework (Sections 3 and 4), experimental setup and evaluation protocols (Section 5), and quantitative/qualitative comparisons (Tables 1–4). Additional implementation details and ablation studies are included in the Appendix to further support reproducibility. While our source code and pretrained models will be released publicly upon acceptance, the information provided in the manuscript and appendix should enable independent researchers to reproduce our findings.

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

# A APPENDIX

## A.1 RELATED WORKS

**Learning-based Adverse Weather Removal**    While single-weather removal excels in precision, it lacks versatility needed to handle real-world situations with multiple and complex weather conditions. To address this, blind or non-blind all-in-one models for adverse weather removal emerged. All-in-one (Li et al., 2020b), one of the earliest non-blind networks, adopts a CNN with task-specific encoders found via NAS. Zhu *et al.* (Zhu et al., 2023) then use a two-stage strategy that learns weather-general and weather-specific priors with a UNet backbone for blind weather removal. Adopting the Transformer (Vaswani, 2017), TransWeather (Valanarasu et al., 2022) matches weather-type

queries to feature keys/values, while AWRCP (Ye et al., 2023) operates in latent space with codebook priors. PromptIR (Potlapalli et al., 2023) introduces plug-in prompt blocks, which use prompts to encode degradation-specific information and a prompt interaction module that dynamically guide the transformer-based restoration network. Adopting the prompt mechanism, Prompt-in-Prompt (PIP) (Li et al., 2023) learns a degradation-aware prompt and a basic restoration prompt, combining them via prompt-to-prompt and selective prompt-to-feature interactions as a plug-in module that improves multi-weather robustness. MiOIR (Kong et al., 2024) learns tasks sequentially, using prompts to reduce interference between them and to stabilize training across diverse weather conditions. U-WADN (Xu et al., 2024) adds a unified-width, nested backbone with an automatic width selector so the model focus on key points, improving efficiency in all-in-one weather removal. UtilityIR (Chen & Pei, 2024) purposes an aware of degradation type and severity model, pairing a marginal quality ranking loss with adaptive normalization/attention to tune restoration strength and handle unseen mixes. OneRestore (Guo et al., 2024) targets composite degradations by fusing scene descriptors with image features, enabling controllable restoration without assuming a single weather type. AWRaCLe Rajagopalan & Patel (2025) uses a degraded-clean context pair as a visual prompt, extracting and fusing weather and type semantics and degradation appearance cues, thus the model perceives the before–after contrast and performs targeted correction.

**VLM-informed Image Restoration** Language-driven image restoration models aim to use natural language to remove the effects of degradation to output a clean high-quality image. In recent years, VLM architectures achieve significant breakthroughs in language-driven image restoration tasks to guide deep learning-based models. TextIR (Bai et al., 2023) leverages CLIP (Radford et al., 2021) to guide restoration by aligning text-based description with features of the restored image. DACLIP (Luo et al., 2023a) presents a degradation-aware VLM trained on text descriptions that guides a SDE-based restoration model to learn high-quality image features. InstructPix2Pix (Brooks et al., 2023) introduces a novel diffusion-based image editing method that can modify specific features in an image based on text-based instructions. InstructIR (Conde et al., 2025) deploys human-written instructions to guide an all-in-one restoration model by feature masking. Co-Instruct (Wu et al., 2025) designs a Large multi-modality model trained on a instruction-based dataset, capable of providing detailed reasoning and answer open ended questions. VLU-Net (Zeng et al., 2025) aims at an interpretable unfolding solver whose gradient step is steered by VLM to auto-select degradation transformers, and with hierarchical feature unfolding, it delivers all-in-one restoration. LDR (Yang et al., 2024) derives a pixel-wise degradation map from natural-language queries to a VLM and uses it to route MoE experts, enabling adaptive restoration without explicit weather labels. InstructRestore (Liu et al., 2025) introduces instruction-guided, region-customized image restoration by building a dataset and a ControlNet-style model which turns natural language region descriptions into masks and per-region feature modulation.

**Diffusion and Flow-based Image Restoration** Modern restoration methods often base their models as stochastic differential equations (SDEs), where the forward process gradually corrupts an image with continuous time-steps of random noise. Reverse-time SDE can also be done to recover the original image. Pioneering SDE-based image restoration, Song *et al.* (Song et al., 2020) proposes a trainable neural network to approximate the score function used for correcting restoration in reverse-time SDE. IR-SDE (Luo et al., 2023b) extends prior works by replacing the standard SDE-based corruption with a mean-reverting process, providing a more true representation of image degradation types. Adopting from these works, denoising diffusion probabilistic models (Ho et al., 2020) introduce a discretized version of the SDE built on Markov chains, leading to a more simplistic and practical framework for image restoration. While SDE-based models are performant in image restoration, their inherent randomness is inefficient, requiring large number of small time-steps for accurate results. To address this, Song *et al.* (Song et al., 2020) further show that reverse-time SDE can be expressed as a probability flow ordinary differential equation (ODE), an efficient alternative that deterministically generates transport mappings via velocity fields to reverse noise. However, ODE-based models are often expensive and slow in practice as most solutions relied on simulating ODE trajectories. Flow matching (FM) (Lipman et al., 2022) is introduced as a simulation-free method that learns transport mappings with a trained neural network. To further build upon FM, Rectified flow (RF) (Liu et al., 2022) is proposed as a more simplistic solution using linear transport paths for noise-data pairs, achieving similar performance at reduced computational cost. Extending from RF, FlowIE (Zhu et al., 2024) uses linear many-to-one transport mappings instead of one-to-one noise-data pairs for strong performance across different restoration tasks. PMRF (Ohayon et al., 2024)

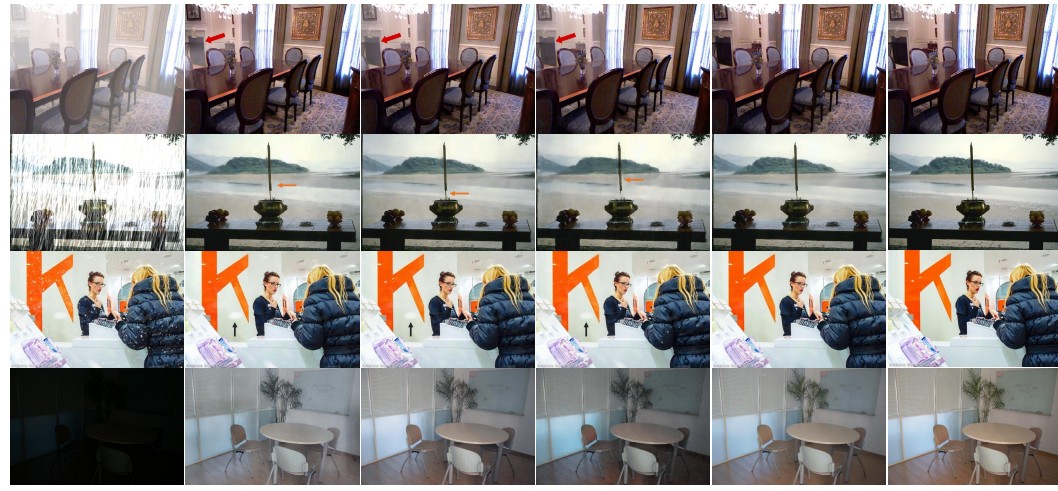

Figure 8: Visual comparisons on setting II. Previous methods often leave residual degradations or introduce artifacts (red, orange, and black arrows), whereas our RectiWeather restores clearer structures, sharper details, and more natural colors, closely matching the ground truth.

Table 6: Ablations on AWR-QA, AMN, and WWA on setting I.

| Configurations | PSNR | SSIM | FID | MUSIQ |
|---|---|---|---|---|
| Full RectiWeather | 30.37 | 0.932 | 10.46 | 63.12 |
| Variant 1 | 28.48 | 0.920 | 13.78 | 62.49 |
| Variant 2 | 29.53 | 0.928 | 11.78 | 62.77 |
| Variant 3 | 29.74 | 0.930 | 11.31 | 62.92 |

uses rectified flow to approximate the optimal transport map, moving posterior mean predictions towards the ground-truth for high-quality image restoration.

## A.2 ABLATION STUDY

To analyze the contribution of each component in our method, we conduct extensive ablation studies on setting I.

**AWR-specific Question Answering, Attribute-Modulated Normalization, and Weather-weighted Adapters** To study the importance of our proposed AWR-specific QA and LPP-Attn, we remove these components from RectiWeather and denote the remaining network as Variant 1. Essentially, vanilla AdaIR serves as the posterior estimator in this setting, and Tab. 6 reports the average quantitative performance of Variant 1 on Setting I. Compared with Variant 1, the full RectiWeather shows significantly improved PSNR and FID by integrating soft VLM perceptions via our Attribute-Modulated Normalization and Weather-Weighted Adapters. In addition, we introduce Variant 2, which employs only the attribute perceptions, and Variant 3, which employs only the weather perceptions. These comparisons demonstrate that the two types of perceptions work complementarily to yield enhanced performance

**Perception-aware Rectified Flow** As reported in Tab. 7, we implement several adaptations to illustrate the importance of the source distribution built upon soft perceptions and the perception-aware velocity estimation. We observe that, if no noise is added to $\mathbf{x}_0$, RF tends to learn a deterministic flow path, while the perceptual metrics do not increase much. When the noise level is fixed, fidelity drops markedly, as evidenced by a heavy decrease in PSNR and SSIM. We observe similar behavior when a vanilla transformer backbone, without the guidance of VLM perceptions, also struggles to find the optimal flow map.

Table 7: Ablations on perception-aware rectified flow on setting I.

| Configurations | PSNR | SSIM | FID | MUSIQ |
|---|---|---|---|---|
| Full RectiWeather | 30.37 | 0.932 | 10.46 | 63.12 |
| No Noise | 29.24 | 0.927 | 14.27 | 61.88 |
| Fix Noise Level (0.1) | 28.27 | 0.920 | 14.55 | 61.75 |
| w/o perceptions in $f_\theta$ | 28.58 | 0.922 | 14.44 | 61.87 |

## A.3 LLM USAGE

Large Language Models (LLMs) were exclusively used for language refinement such as improving grammar, clarity, and flow of the manuscript. Ideas, methodologies, experiments, and analysis were fully carried out by the authors.

