# OpenReview forum: "RectiWeather: Photo-Realistic Adverse Weather Removal via Zero-shot Soft Weather Perception and Rectified Flow"
_ICLR.cc/2026/Conference — ICLR 2026 Conference Withdrawn Submission_

### Official Review · Reviewer_Dyty · 2025-10-30

**Soundness:** 2
**Presentation:** 2
**Contribution:** 2
**Rating:** 4
**Confidence:** 3

**Summary:**

The paper proposes RectiWeather, an all-in-one framework for adverse weather removal. By generating soft weather perceptions through an Adverse Weather Removal Question Answering (AWR-QA) module, the method guides the restoration process using attribute-modulated normalization (AMN) and weather-weighted adapters (WWA). Several baseline restoration models are evaluated to demonstrate the effectiveness of the proposed approach. Experiments are conducted on images degraded by snow, haze, rain, and low-light conditions.

**Strengths:**

The paper is easy to follow, and the proposed components are well validated across different datasets and backbone networks.

**Weaknesses:**

-  Unclear Motivation

One of the stated motivations is to reliably and accurately perceive the underlying weather condition from a degraded input. However, the method simply employs a predefined Vision-Language Model (VLM). Although the AWR-specific Question Answering (AWR-QA) module is introduced, the improvement in perceiving underlying weather conditions appears to stem primarily from the pretrained VLM, rather than from any novel mechanism proposed in this work.

- The paper appears to overstate its contributions.

The only out-of-domain degradation evaluated (Tab. 5) is low-light enhancement, while other tables simply study the same types of degradations using different training and testing datasets. This evidence is insufficient to support the claim of an all-in-one framework, as the degradation types are limited and the model’s generalization ability remains unclear.

From a technical perspective, the proposed attribute-modulated normalization (AMN) seems largely equivalent to adaptive instance normalization (AdaIN). The authors are encouraged to clarify this relationship or highlight any substantive differences.

Regarding the Weather-Weighted Adapters (WWA), the design introduces N separate branches, each corresponding to a specific weather type. This raises concerns about generalization to unseen weather conditions, as the approach appears constrained to the predefined categories used during training.

- Definition–Question–Answering Process

The process of injecting definitions of weather conditions aims to strengthen the Vision-Language Model (VLM) by constraining its decision boundaries. However, this approach resembles a manual optimization process. In a reasoning model (which might be worth to explored with), the reasoning procedure should ideally be automatic rather than manually guided.

**Questions:**

- Producing Soft Perceptions via Multiple Conversations

If the goal is to obtain soft perceptions, it may be more appropriate to utilize the token features before projection to the logits, as they better preserve nuanced information.

- Discussion of the related work

There are other relevant works such as [1]. The authors are encouraged to describe the differences between their method and these related approaches to better highlight the novelty and contributions of their work.

[1] Language-driven all-in-one adverse weather removal.

---

### Official Review · Reviewer_6A1m · 2025-10-31

**Soundness:** 3
**Presentation:** 3
**Contribution:** 2
**Rating:** 4
**Confidence:** 4

**Summary:**

This paper proposes RectiWeather, an all-in-one adverse weather removal framework that leverages zero-shot soft perceptions from Vision–Language Models (VLMs) to enhance generalization and realism. The method introduces an AWR-specific Question Answering (AWR-QA) module to extract weather and attribute cues, which are integrated into restoration networks via Attribute-Modulated Normalization (AMN) and Weather-Weighted Adapters (WWA). A perception-aware rectified flow further refines outputs to improve perceptual quality. Experiments across multiple weather conditions show that RectiWeather achieves state-of-the-art performance in both fidelity and perceptual metrics on in- and out-of-distribution benchmarks.

**Strengths:**

1. Novel integration of zero-shot perception into AWR. The paper introduces a creative use of pretrained vision–language models through the AWR-QA module to derive soft weather and attribute perceptions, enabling effective weather-aware adaptation without task-specific fine-tuning.

2. Well-Integrated Framework. The method combines multiple well-motivated components—AMN, WWA, and perception-aware rectified flow—into a coherent architecture that simultaneously enhances fidelity and perceptual quality, addressing the long-standing trade-off in AWR models.

**Weaknesses:**

1. Limited interpretability of perception-guided mechanisms. The paper lacks detailed analysis of how the extracted soft weather perceptions quantitatively influence restoration behavior, leaving unclear the internal relationship between perception signals and visual quality improvements.

2. Computational efficiency and scalability. The introduction of VLM-based perception and rectified flow modules increases computational overhead, but the paper provides limited quantitative analysis of inference cost, scalability to high-resolution images, or deployment feasibility.

3. Ablation and component validation depth. Although ablation results are presented, the evaluation does not fully isolate the contributions of AMN, WWA, and the rectified flow components under identical training settings, which limits the clarity of their individual effects.

4. Lack of Analysis on Failure Cases and Robustness. The paper only briefly mentions performance drops under unseen haze patterns. A more detailed analysis on failure-case or robustness would strengthen the claims about generalization.

**Questions:**

See the weakness.

---

### Official Review · Reviewer_pqNC · 2025-11-01

**Soundness:** 2
**Presentation:** 3
**Contribution:** 1
**Rating:** 2
**Confidence:** 5

**Summary:**

While the paper introduces an interesting idea of integrating zero-shot weather perception from vision–language models (VLMs) into an all-in-one adverse weather removal framework, it falls short of the standard for a top-tier venue like ICLR. The technical novelty, experimental depth, and positioning relative to recent large-model and diffusion-based restoration methods are not convincing. Fail to show the results on REAL challenge images.

**Strengths:**

- The paper presents a clear motivation and overall framework. The use of vision–language models for weather perception is a meaningful direction that follows the current trend of multimodal intelligence.

- The architecture is described in an organized and understandable way, and the figures effectively illustrate the relationships between the AWR-QA module, the adaptive normalization layers, and the rectified flow.

- The experimental section includes a broad evaluation across several weather degradation types, which demonstrates a reasonable amount of implementation work.

**Weaknesses:**

- The main weakness of the paper lies in the lack of comparison and positioning with the most recent large-model and VLM-based image restoration methods. Recent frameworks proposed in ECCV 2024, ICCV 2025 already explore text-conditioned or perception-guided restoration, but these are not properly discussed or compared. The proposed “zero-shot perception” component is not fundamentally new and is close to common prompt-guided modulation used in vision–language models. The rectified flow stage also follows existing perception-aware flow or diffusion formulations without offering theoretical or algorithmic innovation.

- The technical novelty is limited because the method mostly combines existing modules rather than introducing new principles. The perception extraction is implemented through simple logit-based weighting and does not demonstrate genuine semantic understanding or reasoning capability. The paper also lacks in-depth analysis of how these soft perceptions actually improve image quality or model generalization.

- The experiments rely primarily on synthetic datasets, which do not represent the difficulty of real adverse weather conditions. The evaluation results show only moderate improvements and do not convincingly demonstrate robustness or realism under challenging real-world scenarios. The claim of “photo-realistic” restoration is not well supported by visual evidence, since many results still appear smooth and lack fine texture. There is no perceptual user study or realistic benchmark to confirm visual authenticity.

- Moreover, the paper does not analyze the computational cost or convergence behavior of the rectified flow, and the limitations section admits non-negligible runtime overhead.

**Questions:**

See Weaknesses.

---

### Official Review · Reviewer_cQaz · 2025-11-02

**Soundness:** 3
**Presentation:** 2
**Contribution:** 2
**Rating:** 4
**Confidence:** 4

**Summary:**

This paper proposes RectiWeather, an all-in-one adverse weather removal framework guided by zero-shot soft perceptions from vision-language models (VLMs), aiming to address the limitations of existing methods in poor generalization and lack of photo-realism. The core contributions are threefold: designing an AWR-QA module to extract soft perceptions of weather types and low-level attributes; introducing Attribute-Modulated Normalization (AMN) and Weather-Weighted Adapters (WWA) to integrate perceptual information into baseline models for enhanced degradation awareness; and developing a perception-aware rectified flow model to balance fidelity and perceptual quality. Extensive experiments on multiple weather scenarios and out-of-distribution (OOD) data demonstrate that the proposed method achieves state-of-the-art performance across various metrics, providing a new effective approach for adverse weather image restoration.

**Strengths:**

1. For the first time, zero-shot soft perceptions from VLMs are applied to the AWR task, breaking through the limitations of traditional explicit classification or implicit perception. The definition-question mechanism of the AWR-QA module effectively improves the perceptual robustness in out-of-distribution scenarios.
2.The experimental design is comprehensive, covering multiple weather types, adaptation verification of baseline models, and OOD evaluation. Quantitative and visual results fully support the core claims; component ablation experiments clearly verify the necessity of each module. Moreover, their proposed method achieves state-of-the-art performance across various scenes.

**Weaknesses:**

1. Limited generalization to extreme scenarios: Performance improvement is limited for degradation patterns significantly different from training data (e.g., special haze textures), and the processing effect of extreme weather combinations (such as heavy rain + dense fog; real scenes) is not deeply explored.
2. Suboptimal computational efficiency: The integration of VLM perception and rectified flow model introduces additional runtime overhead. Processing a 1024×1024 image requires an extra 0.4 seconds on a single 5090 GPU, and there is a lack of comparative analysis with lightweight models.

**Questions:**

1.In the AWR-QA module, how are the definition texts of weather types and attributes designed? Have different expression methods been tried, and what is their impact on perception accuracy?
2.What is the basis for setting the noise scale parameters (δ_min=0.025, δ_max=0.1) of the rectified flow model? Will the model performance change significantly if the parameter range is adjusted?
3. What is the model's processing effect on extreme weather combination scenarios (such as heavy rain + dense fog, light snow + low light)? Are there corresponding experimental verifications or optimization strategies?
4.The AWR-QA module relies on pre-trained VLMs. Are there differences in the impact of different VLMs  on perception results and final restoration performance?

---

### Note · Authors · 2025-12-04

I have read and agree with the venue's withdrawal policy on behalf of myself and my co-authors.